# On the Electrochemically Active Surface Area Determination of Electrodeposited Porous Cu 3D Nanostructures

Birutė Serapinienė [ID], Laima Gudavičiūtė [ID], Skirmantė Tutlienė, Asta Grigucevičienė [ID], Algirdas Selskis [ID], Jurga Juodkazytė [ID] and Rimantas Ramanauskas *

Center for Physical Sciences and Technology, Sauletekio 3, LT-10257 Vilnius, Lithuania; birute.serapiniene@ftmc.lt (B.S.); laima.gudaviciute@ftmc.lt (L.G.); skirmante.tutliene@ftmc.lt (S.T.); asta.griguceviciene@ftmc.lt (A.G.); algirdas.selskis@ftmc.lt (A.S.); jurga.juodkazyte@ftmc.lt (J.J.)
* Correspondence: rimantas.ramanauskas@ftmc.lt

**Abstract:** Porous 3D Cu layers with the following average parameters: thickness ~35 μm, pore density ~$4.0 \times 10^6$ cm$^{-2}$, and pore sizes ~25 μm were electrodeposited from an acidic sulphate electrolyte, and the suitability of different electrochemically active surface area determination methods for characterising these electrodes was assessed. Structural characterisation of the samples was conducted using SEM and an optical profiler, while electrochemical measurements were performed using cyclic voltammetry and electrochemical impedance spectroscopy. The evaluation of electrochemically active surface area involved the underpotential deposition of Tl and Pb monolayers as well as double-layer capacitance measurements. The results obtained indicate that both methods yield similar results for non-porous Cu electrodes. However, for Cu 3D nanostructures, the evaluation mode significantly influences the results. Double-layer capacitance measurements show significantly higher values for the electrochemically active surface area compared to the underpotential deposition (UPD) technique. The complex spatial structure of the Cu 3D layer hinders the formation of a continuous monolayer during the UPD process, which is the principal reason for the observed differences.

**Keywords:** 3D copper foam; electrochemically active surface area; surface roughness factor



## 1. Introduction

An exclusive feature of the Cu surface is its capability to electrochemically convert $CO_2$ into hydrocarbons with a significant Faradaic efficiency [1]. The catalytic activity of metal is highly sensitive to electrolysis conditions, including surface structure, morphology, and real surface area ($S_R$). Simple polycrystalline Cu electrodes possess a rather small surface area; therefore, their efficiency in this reaction is low. Meanwhile, high values of $S_R$ are desirable for electrocatalytic applications. Three-dimensional nano-ramified Cu electrodes or foams can be produced by metal electrodeposition accompanied by intensive hydrogen evolution, resulting in plenty of connected and unconnected pores that are evenly distributed within the Cu matrix [2]. Such structures with high $S_R$ values are broadly used in multiple applications, including $CO_2$ reduction [3].

The electrocatalytic performance of porous electrodes depends strongly on the magnitude of their $S_R$, which, in the case of $CO_2$ reduction, seems to affect both reaction activity and selectivity [4]. An accurate estimation of porous electrode $S_R$ is of high importance, as precise knowledge of this parameter is crucial for comparing the behaviour of various catalytic systems. Meanwhile, not all studies related to $CO_2$ reduction on Cu 3D electrodes have paid proper attention to $S_R$ evaluation, whereas others have applied methods whose relevance to the case of porous electrodes is questionable [5].

The aim of this study was to assess the applicability of the known methods used in real surface area determination of Cu electrodes for the characterization of 3D Cu nanostructure.

The active surface area for a particular application depends on the length scale at which the surface plays a role. In applications involving electrochemical reactions, the

electrochemically active surface area ($ES_r$), the area that transfers the charge to species in solution, is the key parameter [6]. It depends on how well the electrolyte accesses the pores and is influenced by the surface roughness. As Cu $ES_r$ values were of particular interest in our study, electrochemical methods for Cu 3D structure analysis were applied. Numerous methods of real surface area determination for various materials have been summarised by Trassati and Petrii [7].

Electrochemical $ES_r$ determination methods are based on two separate approaches. The first one deals with the measurement of charge associated with the deposition or removal of a chemisorbed monolayer of species. For the noble metals, it is usually hydrogen or oxygen, while metal underpotential deposition (UPD), assuming monolayer deposition, can be applied for the larger group of metals, including Cu [5,8–13]. Another method for evaluating $ES_r$ is based on the measurement of the electrical double-layer capacitance of the electrode. This technique is an effective way of determining the wetted area available for electrochemical reactions, which corresponds to $ES_r$.

It is known that Tl, Pb, and Cd can be deposited on Cu surfaces under UPD conditions [14,15] and these processes can be applied for Cu $ES_r$ evaluation [5]. The detailed analysis of Tl and Cd UPD on Cu was performed in our early work [16] with the intention to adjust it for the evaluation of copper $ES_r$. An optimal concentration of Tl+ ions in the solution as well as the conditions for maximum surface coverage were determined, and the suitability of this method for the determination of copper $ES_r$ has been demonstrated. Similar investigations performed with the Cu/Cd UPD system yielded significantly higher $ES_r$ values than the methods applied; therefore, the latter method was suggested to not be suitable.

The method of monolayer formation is claimed to be more sensitive than those based on double-layer charging since the charge consumed in UPD is, as a rule, an order of magnitude higher [7]. However, the latter method possesses several restrictions related to the fact that the identification of the end point for the metal adsorption may be uncertain, the surface distribution of the UPD species may be unknown, the adatom deposition may occur with partial charge transfer, and the new phase formation may result in more condensed monolayers, multilayers, or cluster growth [7]. Despite the mentioned possible limitations, the UPD method is widely used for $ES_r$ evaluation of Cu samples. Meanwhile, recently, the Pb UPD process was applied for $ES_r$ evaluation of non-porous [11,16–18], and porous Cu [5] structures more frequently than Tl UPD [14,15].

Monolayer oxidation of Cu in an alkaline solution was recently used by different scholars for $ES_r$ determination of porous and non-porous Cu electrodes [3,5]. This method is applicable to metals showing distinct regions of oxide monolayer formation and reduction [7], which is not always the case for Cu samples [16]. The oxidation of Cu in alkaline media under applied potential results in the formation of different oxides and hydroxides. During an oxidation scan, the first current peak corresponds to the formation of $Cu_2O$, and according to [5], for the small range of OH⁻ molarity (0.1–1 M) and scan rate (50–100 mVs$^{-1}$), it can be assumed that the oxidation of the Cu surface leads to the formation of a monolayer film. However, our previous studies have indicated that under the mentioned conditions, $Cu_2O$ formation on the metal surface does not end with monolayer formation; in addition, this process is irreversible, which causes surface morphology changes and therefore is not suitable for Cu $ES_r$ determination [16].

The surface area of the porous metal structures strongly depends on the manufacturing method. Up to date, the quantitative information on the $ES_r$ of this type of porous electrode is contradictory, as the determined values of the surface roughness, $f_R$ (ratio between $ES_r$ and geometrical area of the sample $S_g$), vary between 200 and 800 [14,19,20], depending on the method applied. It is evident that there is no consensus among the investigators on what method of $ES_r$ determination is most suitable for porous Cu nanostructures.

In this study, electrochemical methods employing cyclic voltammetry (CV) and electrochemical impedance spectroscopy (EIS) were applied for $ES_r$ evaluation of Cu 3D structures,

while $S_R$ values of the metal substrate surface, which was covered with a Cu 3D layer, were additionally evaluated using optical profilometry.

## 2. Materials and Methods

### 2.1. Sample Preparation

A polycrystalline copper foil of 1 cm$^2$ geometric area was used as a substrate for sample preparation. Firstly, a Cu layer of 8 μm thickness was deposited from the acidic sulphate solution to prepare a "plain" electrode with a surface roughness index of $f_R$~2 [14]. Secondly, Cu 3D structures were electrodeposited on the plain electrode. The electrolyte compositions and electrodeposition parameters are listed in Table 1. Analytical-grade chemicals and deionised water were used for the preparation of all electrolytes applied.

**Table 1.** Composition of the Cu deposition electrolytes and plating conditions.

| Cu Layer | Electrolyte | Current Density | Deposition Time |
|----------|-------------|-----------------|-----------------|
| Plain | 0.5 M Cu$_2$SO$_4$ <br> 0.5 M H$_2$SO$_4$ <br> 1 M C$_2$H$_5$OH | 0.02 A·cm$^{-2}$ | 18 min |
| Foam | 0.2 M Cu$_2$SO$_4$ <br> 1.5 M H$_2$SO$_4$ | 3.0 A·cm$^{-2}$ | 20 s |

### 2.2. Electrochemical Measurements

All electrochemical measurements were carried out at ambient temperature in a three-electrode electrochemical cell using a Pt counter electrode, Ag/AgCl or Hg/Hg$_2$SO$_4$ reference electrodes, and a potentiostat/galvanostat AUTOLAB 302. All electrolytes were purged with Ar gas for no less than 20 min prior to measurements. All potentials in the text are reported versus a standard hydrogen electrode (SHE).

Tl and Pb UPD measurements were performed using 1 M Na$_2$SO$_4$ + 0.5 mM Tl$_2$SO$_4$ and 0.01 M HClO$_4$ + 1 mM PbCl$_2$ electrolytes, respectively. Metal monolayers were formed at a potential value (E) 10 mV more positive than thermodynamically possible metal deposition potentials. The UPD deposition time of both metals in the case of the plain Cu electrode was set to 200 s in accordance with the previous studies [16], while for Cu 3D structures, the maximum surface coverage state by Tl and Pb monolayers was determined to be no less than 900 s. In the next step, the formed UPD layers were anodically dissolved at a potential scan rate of 10 mV·s$^{-1}$. The amount of charge consumed for anodic dissolution of Tl or Pb monolayers, $Q_a$, was calculated by integrating the areas under the anodic current peaks according to the following equation:

$$Q_a = \frac{1}{\nu} \int_{E_2}^{E_1} j \mathrm{d}E, \tag{1}$$

where $\nu$ is scan rate (V·s$^{-1}$), $j$ is current density (μA·cm$^{-2}$), and $E$ is applied potential (V) [17]. All current density values mentioned in the text refer to the geometric area of the samples.

In order to calculate the surface roughness factor ($f_R$), Equation (2) was applied [14]. The theoretical amounts of charge ($Q_{Tl}$ and $Q_{Pb}$) corresponding to a monolayer of Tl or Pb on 1 cm$^2$ of Cu surface used in the calculations were 112 μC cm$^{-2}$ [14] and 250 μC cm$^{-2}$ [21], respectively.

$$f_R = \frac{Q_a}{Q_{Pb(Tl)}} \tag{2}$$

The double-layer capacitance measurements were performed in a 0.1 M NaOH electrolyte. Cyclic voltammograms (CV) were recorded at different scan rates in non-Faradaic regions, which were set to be −0.5 V–−0.4 V and −0.57 V–−0.42 V for plain and 3D electrodes, respectively. The double-layer capacitance values were determined by plotting the

capacitive current values obtained at $-0.45$ V for the plain electrode and $-0.50$ V for the 3D electrode against the scan rate. The slope of the resulting linear relationship provided the double-layer capacitance value (*C*). $ES_r$ and $f_R$ values were calculated according to the equations:

$$ES_r = \frac{C}{C_{sp}}, \tag{3}$$

$$f_R = \frac{ES_r}{S_G}, \tag{4}$$

where $ES_r$ is the electrochemically active real surface area, cm$^2$; *C* is the copper electrode double-layer capacitance, mF; $C_{sp}$ is the specific double-layer capacitance of copper in an alkaline solution of 0.02 mFcm$^{-2}$ [3]; $f_R$ is the surface roughness factor; and $S_G$ is the geometrical surface area of an electrode, cm$^2$.

EIS measurements were performed at the open circuit potential with the FRA2 module applying a signal of 10 mV amplitude in the frequency range of 20 kHz to 0.005 Hz. The data obtained were fitted and analysed using the EQUIVCRT programme of Boukamp [22].

All electrochemical experiments were performed at least in triplicate.

### 2.3. Morphological Characterisation

A two-beam system, Helios Nanolab 650 (FEI), was used in the secondary electron mode at an accelerating voltage of 3 kV to study the morphology of the coatings. The Cu 3D layer porosity was evaluated by applying visual analysis to SEM images.

The surface roughness and morphology of the samples were additionally evaluated using the 3D optical profiler Contour GT-K (Bruker Nano GmbH, Berlin, Germany) in non-contact mode by white light and phase shift interferometry. Surface area measurements were performed using 50× optics to scan an area of 500 × 500 μm. The surface characteristics, including surface roughness, were evaluated based on these measurements. Data acquisition and surface analyses were conducted using Vision64 software.

## 3. Results

### 3.1. SEM and Optical Profilometry

To assess the applicability of real surface area determination methods for porous Cu 3D nanostructures (foams), a specific experimental design was developed. The initial Cu electrode, with the known $f_R$ value ($f_R$~2.2) [14], which is referred to as the "plain" one, was used as a basis for Cu 3D structure electrodeposition. Nonporous Cu surfaces can be examined using various methods, including physical and electrochemical, and these methods usually yield very similar results. The surface roughness and topography of the plain sample, which was a Cu layer electrodeposited from an acidic sulphate solution, were evaluated using a 3D optical profiler in non-contact mode by white light and phase shift interferometry. The obtained optical image of the plain sample is presented in Figure 1a, while the applied software yielded $f_R$~2.21, which was very close to the value indicated above.

The porous 3D Cu electrode, which was used as the research object for $ES_r$ evaluation, was obtained by Cu electrodeposition under extremely high cathodic current density (up to 3 A cm$^{-2}$) in an electrolyte with high acidity. It is proven that the formation of the foam structure is caused by the competitive reaction of Cu deposition and hydrogen evolution, resulting in the formation of a 3D morphology with a very unique pore size distribution and highly porous ramified (dendritic) walls [23,24]. The surface pore size, wall width, and foam thickness depend on the conditions of electrodeposition and the electrolysis regime [25]. SEM images of the surface morphology of the deposited Cu 3D electrode, as well as a cross section of it, are presented in Figure 2.

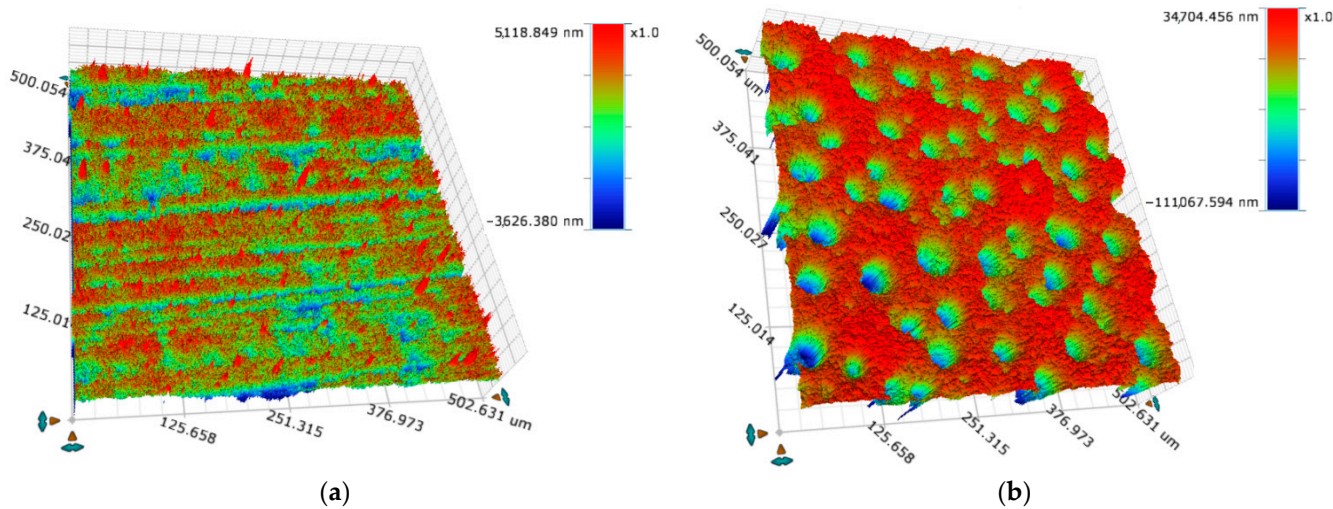

(**a**)      (**b**)

**Figure 1.** Optical images of the surface topography of the plain Cu (**a**) and Cu 3D structure (**b**).

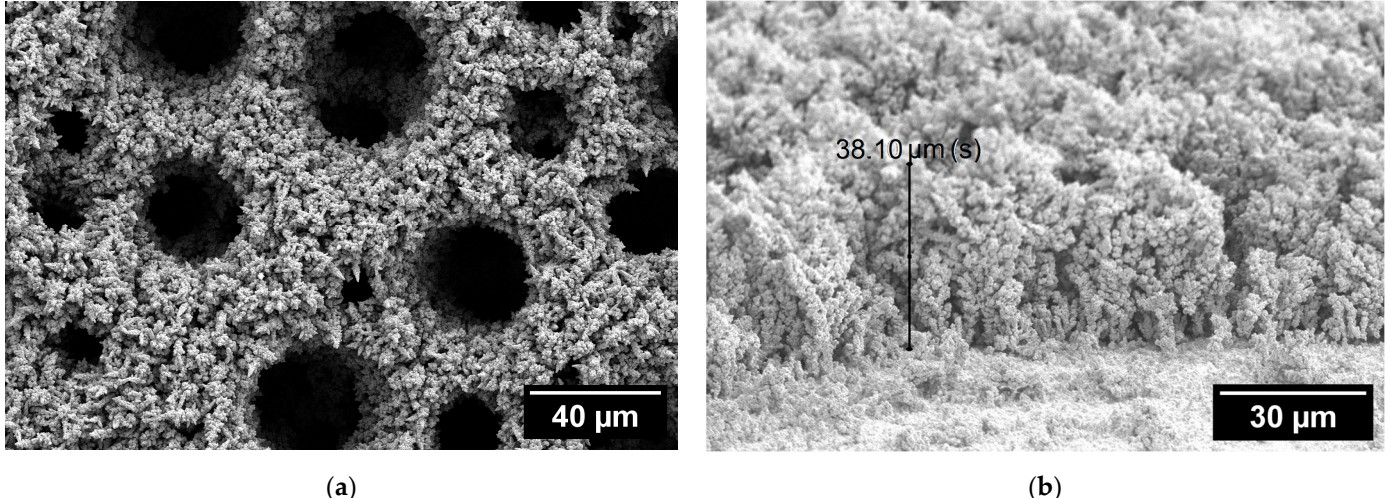

(**a**)      (**b**)

**Figure 2.** SEM images of Cu 3D electrode morphology (**a**) and its cross section (**b**).

Optical profiling and SEM images yielded information on the deposited 3D Cu structure, indicating that the average parameters were pore density $\sim 4.0 \times 10^6$ cm$^{-2}$, pore size $\sim 25$ μm, and thickness $\sim 35$ μm. It is evident that the three-dimensional foam provides a large electrochemically active surface area and that 3D optical profilometry can be applied only for the evaluation of the density of pores (Figure 1b). The latter measurements yielded the average $f_R \sim 7.9$ value for the Cu 3D electrode, which is evidently an inadequate one.

### 3.2. UPD Measurements

The application of UPD processes for $ES_r$ evaluation of Cu samples was initiated with the studies on plain electrodes. The cyclic voltammograms representing Tl and Pb UPD deposition/dissolution processes on plain Cu electrodes are shown in Figures 3 and 4 (black curves), respectively. Integration of the anodic charge under the current peaks in the potential ranges between $-0.30$ V and $-0.130$ V for Tl and $-0.060$ V and $-0.020$ V for Pb yielded information on the $Q_a$ and consequently on the $f_R$ values of the plain (initial or standard) Cu electrode. Both $f_R$ values of $1.8 \pm 0.1$ and $2.5 \pm 0.3$ obtained by Tl and Pb UPD measurements, respectively, are close to the standard $\sim 2.2$ value determined by other authors [14], indicating the suitability of these methods for $ES_r$ evaluation of non-porous Cu.

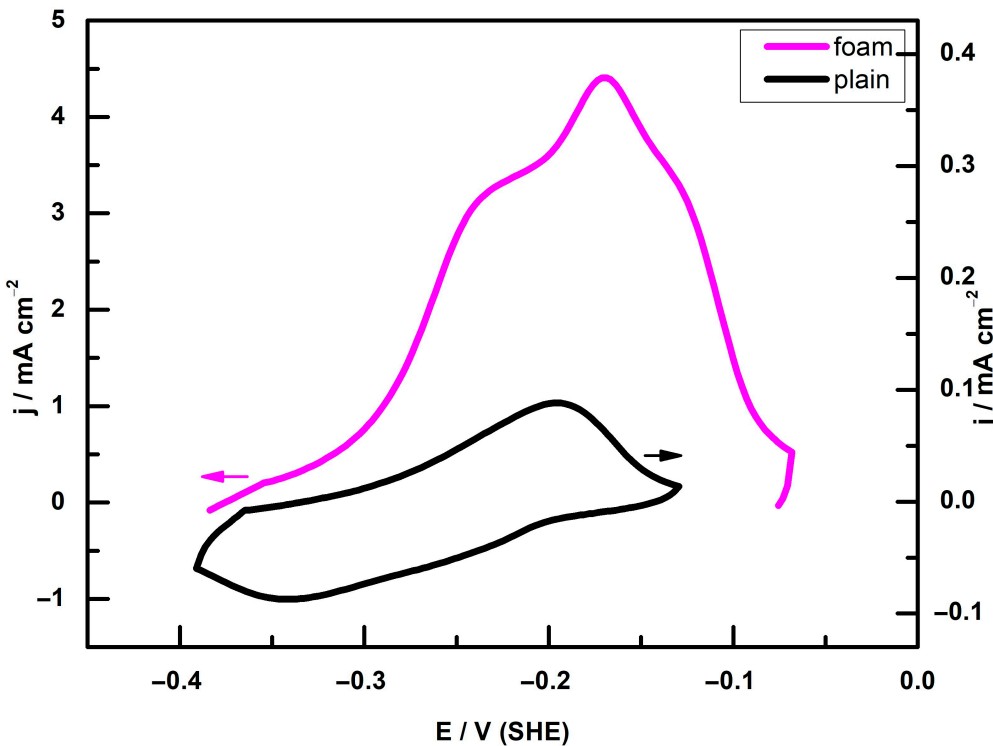

**Figure 3.** Cyclic voltammograms of Tl UPD deposition/dissolution of plain Cu (black curve) and anodic striping of Tl UPD layer from 3D Cu electrode (pink curve) in 1 M $Na_2SO_4$ + 0.5 mM $Tl_2SO_4$ electrolyte, potential scan rate $v$—10 mV s$^{-1}$. Arrows in the graph shows to which axis it belongs to.

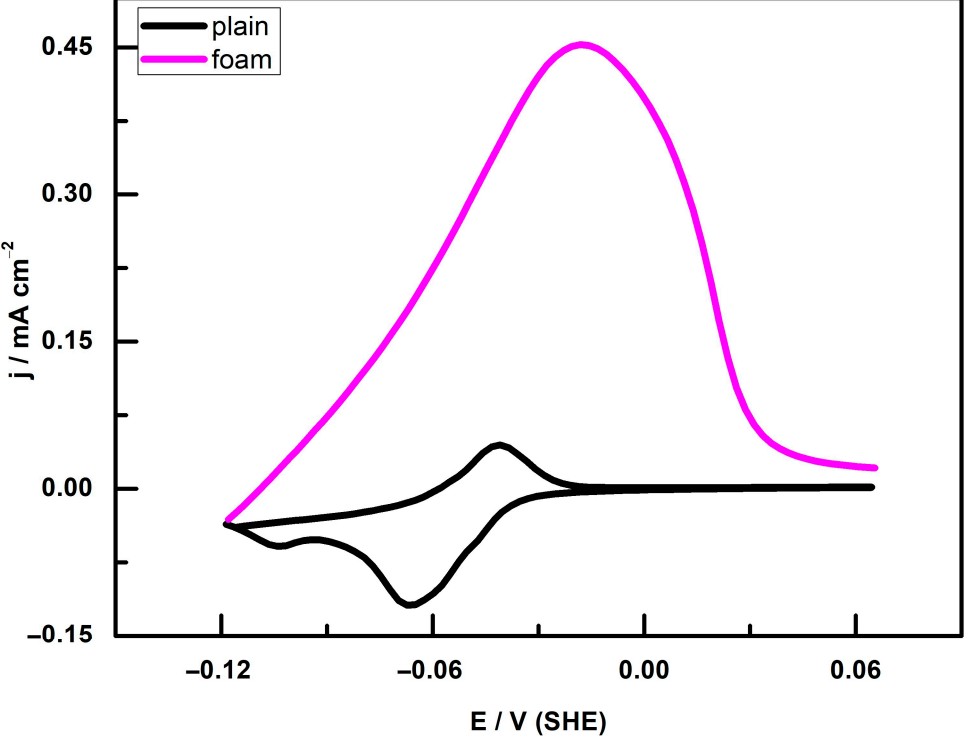

**Figure 4.** Cyclic voltammograms of Pb UPD deposition/dissolution on plain Cu (black curve) and anodic striping of Pb UPD layer from 3D Cu electrodes (pink curve) in 0.01 M $HClO_4$ + 1 mM $PbCl_2$ electrolyte. Potential scan rate $v$—10 mV s$^{-1}$.

The presence of a porous Cu structure, which results in a large surface area, can potentially cause additional hindrances for the adsorption of Pb and Tl on such complex spatial surfaces. Hence, it was logical to determine the conditions that yield the maximum surface coverage of Tl and Pb underpotential deposition (UPD) layers by applying potentiostatic deposition conditions. This was achieved by selecting the highest charge values that corresponded to the dissolution of adsorbed Tl and Pb ions on the porous Cu electrode. Experiments revealed that the maximum surface coverage for both metals was achieved when the duration of the underpotential deposition (UPD) process was equal to or longer than 900 s (Figure 5).

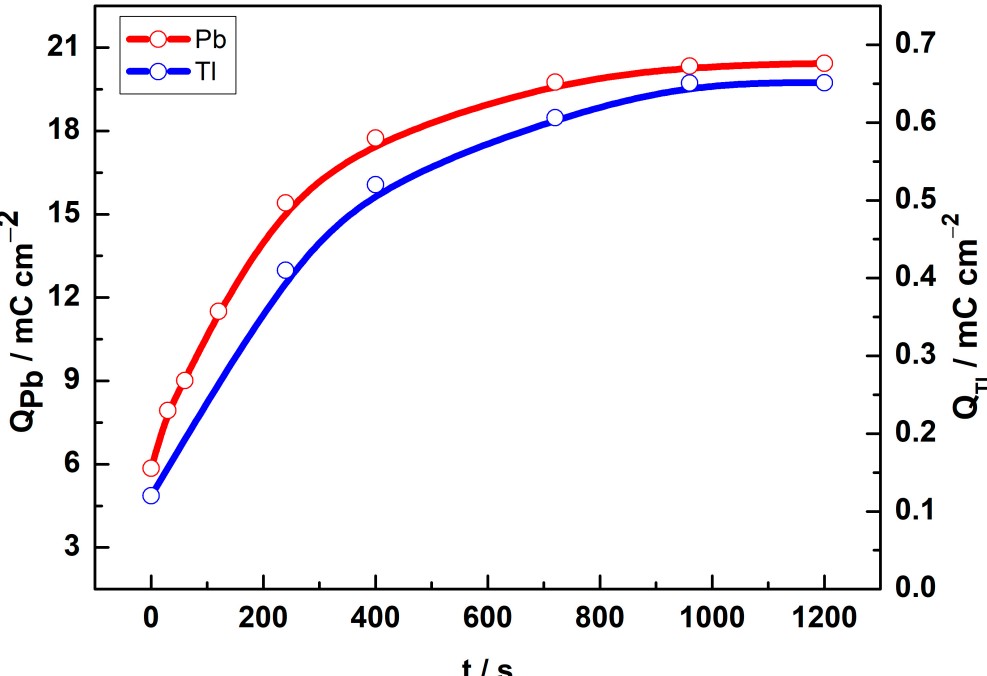

**Figure 5.** Influence of UPD process duration on the charge required to anodically dissolve the UPD layer: red—Pb UPD and blue—Tl UPD.

The main challenge in calculating the charge corresponding to the deposited metal lies in accurately correcting for background processes such as double-layer charging and properly identifying the potential at which the formation or dissolution of a monolayer of metal atoms is completed. The shape of the anodic dissolution peaks observed for Tl and Pb UPD layers on 3D Cu electrodes is relatively simple and consistently yields repeatable results. The only slight difference in the shape of anodic curves is that one for Tl is not fully symmetrical as that corresponding Pb UPD (Figure 3). This phenomenon may be attributed to the varying bonding strength of Tl on different crystal planes of Cu. However, it does not have any significant impact on the final results obtained. The potential ranges where the metal monolayer dissolution takes place are quite evident. The anodic charge within the potential ranges of −0.370 to −0.070 V for Tl and −0.110 to −0.023 V for Pb was evaluated using the methods described in the Experimental Section for determining Cu $ES_r$. The resulting $f_R$ values for the 3D Cu structures are listed in Table 2. As mentioned previously [14], it is widely acknowledged that the $f_R$ values are influenced by the measurement method employed as well as the operational conditions. Similar observations can be made based on the results obtained from the investigation of Cu 3D structures reported herein. The obtained $f_R$ values for Cu 3D samples using the Pb UPD method (around 80 ± 2) and the Tl UPD method (around 116 ± 2) suggest that the $ES_r$ values for Cu 3D structures are lower when using the Pb UPD reaction compared to Tl. This difference can be attributed to the slower kinetics of Pb UPD on the Cu surface, as

previously stated [19]. Additionally, the presence of a porous Cu 3D structure can influence the maximum surface coverage state in the case of Pb UPD.

**Table 2.** Surface roughness factors of Cu electrodes evaluated by different methods.

| Method | Surface | Surface Roughness Factor |
|---|---|---|
| Pb UPD | Plain | $2.5 \pm 0.3$ |
| | Foam | $80 \pm 2$ |
| Tl UPD | Plain | $1.8 \pm 0.1$ |
| | Foam | $116 \pm 2$ |
| Double-layer capacitanceCyclic voltammetry | Plain | $1.8 \pm 0.1$ |
| | Foam | $814 \pm 29$ |
| Double-layer capacitanceImpedance | Plain | $2.1 \pm 0.2$ |
| | Foam | $986 \pm 36$ |
| Optical profiler | Plain | 2.2 |

### 3.3. Double-Layer Capacitance

Double-layer capacitance is influenced by the surface area of the electrode. Consequently, measuring this electrochemical parameter can serve as a means to estimate the real surface area of solid metal electrodes [4]. The electrochemically active surface area of an electrode can be estimated by means of double-layer capacitance measurements using cyclic voltammetry and EIS. Usually, CV curves are recorded in the double-layer charging region at various scan rates. The shape of the curves for plain and 3D Cu electrodes is very similar; therefore, Figure 6a depicts only CV curves for the latter electrode obtained in 0.1 M NaOH solution in the non-Faradaic region ($-0.57$ V$-$$-0.42$ V SHE). The obtained capacitive current values were plotted against the scan rate in Figure 6b, and the slope of the obtained linear dependence yielded the double-layer capacitance values, which were $23.8 \pm 2$ μF and $10.4 \pm 0.58$ mF for the plain and 3D Cu electrodes, respectively. $ES_r$ and $f_R$ values were calculated according to Equations (3) and (4) and are listed in Table 2. It can be observed that for the plain Cu electrode, CV-based double-layer capacitance measurements yielded an $f_R$ value of $1.8 \pm 0.1$, which is very close to the one determined by the UPD method. In contrast, for Cu 3D electrodes, the double-layer capacitance measurements resulted in significantly higher $f_R$ values of $814 \pm 29$ compared to the values obtained through UPD measurements which ranged between $80 \pm 2$ and $116 \pm 2$ (Table 2).

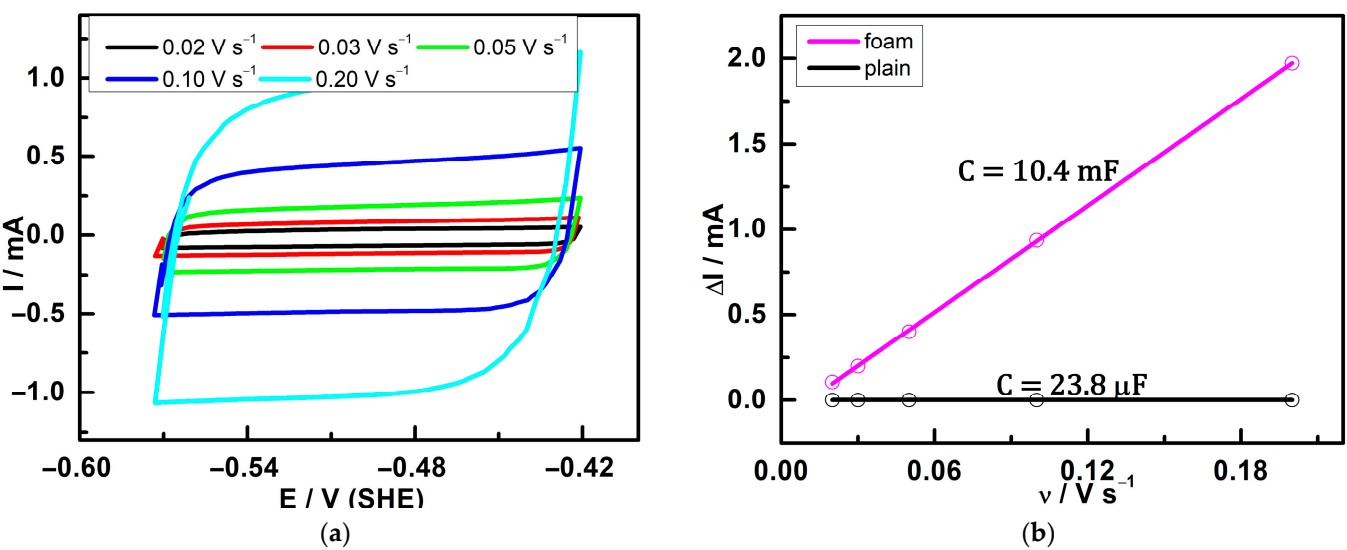

**Figure 6.** (**a**) Voltammograms of Cu foam in the non-Faradaic region recorded at different scan rates and (**b**) linear dependence of change in current on scan rate.

*3.4. EIS Study*

The validity of the double-layer capacitance results obtained by CV was verified by EIS measurements. EIS spectra as Nyquist and Bode plots of the plain and Cu 3D samples recorded in a 0.1 M NaOH solution are presented in Figure 7. A single semicircle is evident in the Nyquist plot of the plain Cu electrode (Figure 7A). Two different equivalent circuit models (Figure 8a,b), which have been widely used for the analysis of the impedance spectra of Cu and Cu porous electrodes [20,26], were applied. The circuit model utilised for the plain Cu electrode (Figure 8a) encompasses the interaction of three components: the solution resistance ($R_S$) in series with a combination of the electrode resistance to the Faradaic process ($R_{ct}$) in parallel with the constant phase element ($CPE_{dl}$), which represents the double-layer capacitance. In contrast, the Nyquist plots of the Cu 3D electrode exhibit two semicircles. In Bode plots, the region between $10^2$ and $10^4$ Hz provides information on the Cu 3D layer parameters, while the low frequency region can be assigned to the charge transfer process (Figure 7B). The equivalent circuit employed for the Cu 3D electrode, as shown in Figure 8b, consists of a series resistance $R_S$ (representing the electrolyte resistance), resistance $R_c$, and a parallel combination of the constant phase element $CPE_c$ (representing transport properties within the electrodeposited Cu 3D layer) and another parallel pair comprising of resistance $R_{ct}$ and $CPE_{dl}$ (associated with the charge transfer reactions) [21]. The chosen equivalent circuits demonstrated good agreement between the experimental data and the simulated data.

For the data fitting, all the capacitances in the equivalent circuits had to be replaced by constant phase elements ($CPE$) [27] to adapt to non-ideal behaviour. $CPE$ instead of capacitors was used in the equivalent circuit models to account for the inhomogeneous properties of the layers. $CPE$ is defined by the admittance $Y$ and the power index number $n$: $Y = Y_o (jw)^n$. The term $n$ shows how far the interface is from an ideal capacitor. $Y_0(CPE_c)$ and $Y_0(CPE_{dl})$ become $C_c$ and $C_{dl}$ for $n = 1$. Table 3 shows that the term $n(CPE)$ for the Cu samples plain and foam has a value > 0.5, and initially it is close to one, suggesting a capacitive response from the electrolyte/coating interface.

**Table 3.** EIS parameters obtained for plain Cu and Cu foam using equivalent circuits shown in Figure 8.

| Sample | $R_s$, $\Omega$ | $R_c$, $\Omega$ | $Y_0(CPE_c)$/ $10^{-6}$ $\Omega^{-1} s^n$ | $n$ $(CPE_c)$ | $R_{ct}$, $\Omega$ | $Y_0(CPE_{dl})$/ $10^{-3}$ $\Omega^{-1} s^n$ | $n$ $(CPE_{dl})$ |
|---|---|---|---|---|---|---|---|
| Plain | 13 | - | - | - | 23,000 | $22.4 \times 10^{-3}$ | 0.8 |
| Foam | 5 | 2.1 | 214 | 0.89 | 471 | 12 | 0.79 |

By applying the equivalent circuits to fit the impedance spectra, a set of fitting parameters was obtained, which is presented in Table 3. The results obtained demonstrate that the values of $R_{ct}$ of the investigated samples were in the range of 20–23 k$\Omega$ for a plain Cu electrode and within 470–600 $\Omega$ for a Cu 3D electrode. The double-layer capacitance $CPE_{dl}$ is found to be about 22 µF for the plain Cu electrode and about 12 mF for the Cu foam electrode, which is comparable to the values obtained from CV-based double-layer capacity measurements (23.8 µF for the plain Cu electrode and 10.4 mF for the Cu foam). Consequently, EIS measurements yielded $f_R$ values of 2.1 ± 0.2 for the plain and 986 ± 36 for Cu 3D structured electrodes. The results obtained suggest that in the case of plain or non-porous Cu electrodes, the electrochemically active surface area values are not affected by the determination method. However, for Cu 3D structures, this parameter shows significant dependence on the evaluation mode. Double-layer capacitance measurements result in considerably higher $ES_r$ values compared to the UPD technique.

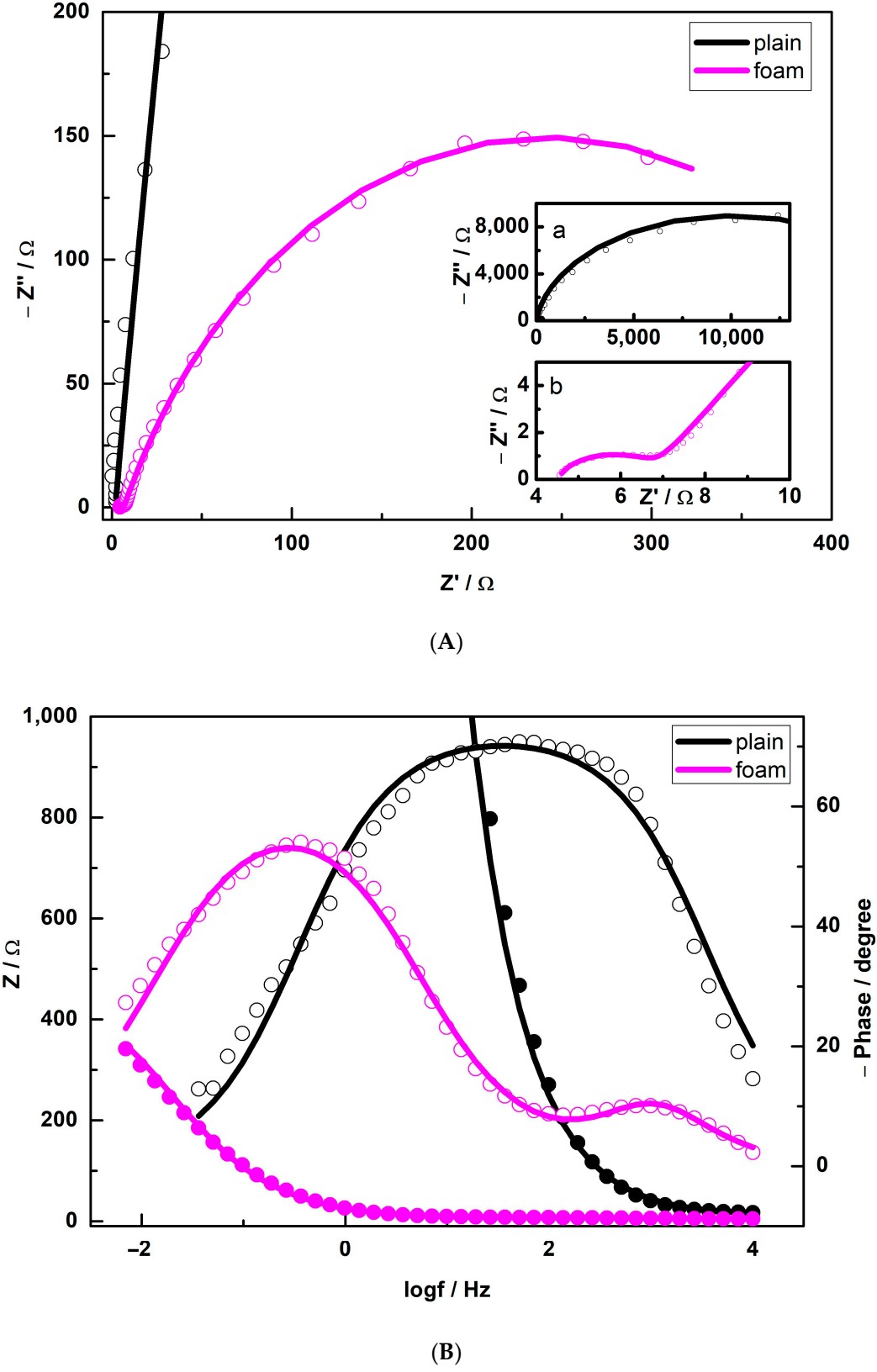

**Figure 7.** EIS data recorded in 0.1 M NaOH solution for plain Cu and foam Cu as Nyquist (**A**) and Bode (**B**) plots. The insets (**a**,**b**) in (**A**) correspond to the Nyquist plots at low and high frequencies, respectively.

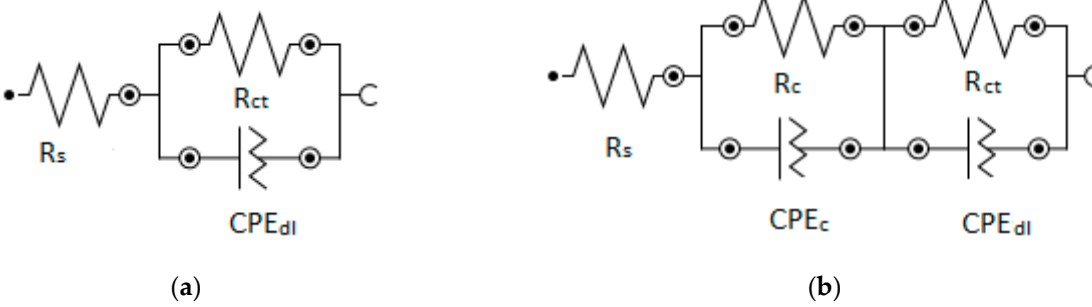

**Figure 8.** The circuit model used for the plain Cu electrode (**a**) and for the Cu 3D electrode (**b**) for EIS data fitting.

*3.5. Suitability of $ES_r$ Determination Methods for Cu 3D Structures*

The results obtained suggest that in the case of plain or non-porous Cu electrodes, the electrochemically active surface area values are not affected by the determination method applied in this study. However, for Cu 3D structures, this parameter shows significant dependence on the evaluation mode. Double-layer capacitance measurements result in considerably higher $ES_r$ values compared to the UPD technique.

In order to establish the possible reasons for this phenomenon, additional measurements were carried out. The pore sizes and wall structures of the foams are tunable by adjusting the deposition conditions [23]. Considering this, Cu 3D samples were produced by varying the deposition time from 5 up to 25 s and were analysed by applying Tl UPD and voltametric double-layer capacitance measurements. The surface morphology of the deposited Cu foams, except for pore density and size, does not differ significantly from that of the sample presented in Figure 2a and therefore is not shown here. The surface roughness factors ($f_R$) and porosity parameters of electrodeposited Cu 3D samples are listed in Table 4.

**Table 4.** Surface roughness factors ($f_R$) and porosity parameters of Cu 3D electrodes deposited at a 3 A cm$^{-2}$ current density and various deposition durations.

| Deposition Duration, s | Average Pore Size, μm | Average Pore Density, cm$^{-2}$ | $f_R$, by Double-Layer Capacitance | $f_R$, by Tl UPD |
|---|---|---|---|---|
| 5 | 7.3 | $3.2 \times 10^7$ | $90 \pm 12$ | $24 \pm 4$ |
| 10 | 16.5 | $1.2 \times 10^7$ | $290 \pm 19$ | $40 \pm 9$ |
| 15 | 21.5 | $8.0 \times 10^6$ | $600 \pm 26$ | $76 \pm 10$ |
| 20 | 25.3 | $4.0 \times 10^6$ | $814 \pm 29$ | $116 \pm 12$ |
| 25 | 28.8 | $3.0 \times 10^6$ | $900 \pm 38$ | $126 \pm 13$ |

It can be observed that the increase in deposition time resulted in an increase in the size and a reduction in the density of pores in the Cu 3D layer. The surface area of this type of electrode is determined by the number and size of holes formed by detached hydrogen bubbles as well as by the wall width between them [23]. Application of both $ES_r$ determination methods revealed an increase in the $f_R$ value of Cu foam with the increase in deposition duration; meanwhile, the double-layer capacitance measurements yielded higher $f_R$ values compared to UPD measurements for all investigated samples.

The more detailed studies of foam formation have revealed that two types of pores are formed by the hydrogen evolution reaction during the electrodeposition process [28]. The first type is macropores (or holes) formed by detached hydrogen bubbles, while the origin of the second type of pores is hydrogen bubbles generated at the tops of the agglomerates of Cu grains during the growth process. In addition, the pore size of foam structures increases

with distance from the substrate [28]. The analysis of the origin of the pores forming during Cu foam electrodeposition was out of the scope of this study; however, it can be supposed that with the growth of the Cu 3D layer, the inner or spatial structure of it becomes more and more complicated, which may be the principal reason for the $f_R$ variation observed when applying different $ES_r$ determination methods. In addition, a minimal three-fold difference between $f_R$ values was observed for the samples deposited at 5 s, while for all the successive ones, a more than seven-fold difference in $f_R$ was found. The latter issues are probably caused by some limitations of the UPD-based $ES_r$ determination method.

Despite the fact that the maximum surface coverage of the porous Cu electrode by Tl and Pb adatoms in the UPD layer was achieved (Figure 5), it was presumed that the formation of a continuous and integral monolayer did not occur due to the complex spatial structure of the samples. Consequently, significantly lower $f_R$ values for Cu foams were obtained. Therefore, double-layer capacitance measurements are recommended for the evaluation of the electrochemically active surface area of porous Cu 3D structures.

## 4. Conclusions

The determination of the electrochemically active surface area, a crucial parameter in electrocatalysis, shows significant dependence on the technique employed. The analysis of non-porous Cu electrodes using various electrochemical or optical methods consistently produces similar results for both real and electrochemically active surface areas. However, a different situation was observed when studying Cu 3D porous structures deposited from acidic sulphate solutions under intensive hydrogen evolution and having intermittent pore size distribution along with dendritic walls and a seemingly large surface area.

Comparing different techniques, the underpotential deposition of Tl and Pb monolayers on the porous Cu 3D electrodes results in significantly lower values of electrochemically active surface area when compared to measurements of double-layer capacitance, which were carried out using voltammetry or electrochemical impedance spectroscopy. It appears that the complex spatial structure of the Cu electrode may account for the observed differences in surface area evaluations. Among the techniques considered, the evaluation of double-layer capacitance through voltametric measurements emerges as the most suitable, owing to its simplicity.

**Author Contributions:** Conceptualisation, R.R.; data curation, B.S., L.G., R.R. and J.J.; formal analysis, B.S., L.G. and R.R.; investigation, B.S., L.G., A.G., S.T. and A.S.; methodology, B.S., L.G. and R.R.; software, B.S. and L.G. supervision, R.R.; writing—original draft, R.R.; writing—review and editing, R.R. and J.J. All authors have read and agreed to the published version of the manuscript.

**Funding:** This research received no external funding.

**Institutional Review Board Statement:** Not applicable.

**Informed Consent Statement:** Not applicable.

**Data Availability Statement:** The data presented in this study are available on request from the corresponding author.

**Conflicts of Interest:** The authors declare no conflict of interest.

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
