# Peer review of "On the Electrochemically Active Surface Area Determination of Electrodeposited Porous Cu 3D Nanostructures"

_coatings, doi:10.3390/coatings13081335_

Round 1

Reviewer 1 Report

Reviewers Comments (Manuscript ID# coatings-2478451.)

The manuscript titled “On the electrochemically active surface area determination of 2 electrodeposited porous Cu 3D nanostructures” by Serapinienė etal, reports the electrodeposition of porous copper under different conditions to attain different morphology. However, the results are not well characterized, and very limited techniques are used to study applications. Porosity, pore volume, oxidation state, and phase nothing are determined. Moreover, the author claims their previous work as reference 16 (published in Chemija in 1990), but not a single author is there in that published work.

In short, due to the lack of novelty and limited studies, I would not recommend the paper to be published in Coatings

Author Response

Expressing our thanks for the comments and suggestions of the Reviewer 1 we would like to give the following explanations.

Comment 1. The manuscript titled “On the electrochemically active surface area determination of 2 electrodeposited porous Cu 3D nanostructures” by Serapinienė et.al, reports the electrodeposition of porous copper under different conditions to attain different morphology.

Response:  Our aim was to assess the applicability of the real surface area determination methods for the nanostructured and porous Cu electrodes, having in mind that the literature data on this topic are contradictive (e. g. references [14,20, 21] in the manuscript). With this intension the porous Cu electrodes were electrodeposited under the certain conditions and were characterized. Electrochemically active surface area (ESr) values were determined applying two different approaches. There was no attempt to deposit Cu 3D structures with the different morphology in the, however, in the reviewed version we have included data on ESr of samples with different porosity. 

Comment 2. However, the results are not well characterized, and very limited techniques are used to study applications. Porosity, pore volume, oxidation state, and phase nothing are determined.

Response: The focus of this work was the applicability of ESr determination methods for the porous Cu electrode. The research object was electrodeposited Cu foam sample, which was characterized applying SEM and optical profiler for morphology examination, while electrochemical techniques were applied for ESr evaluation. Generally, we have used additional techniques, e.g.  XRD measurements (XRD diffractogram is presented below):

Fig. XRD patterns of Cu 3 D samples deposited for 10 (red) and 20 (blue) seconds.

 and we have in our disposition information on the phase composition and crystallite size of investigated samples. However, we decided not to employ them in this work, as our aim was different. These results will be introduced in the following our publications. Due to experimental restrictions (insufficient amount of the research object) we were unable to apply directly Brunauer- Emmett-Teller (BET) method for the surface area determination of the deposited Cu 3D sample. Nevertheless, Cu foam was scratched from the base metal and the resulting powder was analysed. The following results were obtained i.e. surface area = 7.917 m²/g (Correlation coeff. r = 0.999734, C constant = 6.9519), however these results were not included as well, as the electrochemical active surface area was our target.

Comment 3. Moreover, the author claims their previous work as reference 16 (published in Chemija in 1990), but not a single author is there in that published work.

Response. The corresponding authors of the manuscript R. Ramanauskas is the first authors of the reference [16].

Reviewer 2 Report

The paper describes the determination of properties of 3D nanostructured Cu obtained by electrodeposition. The paper is well-written and interesting, however I have few notes:

-       -   How is it possible to regulate the porosity of the metal foam? Which parameter of electrodeposition affects it?

-       -   The dependence of the pore size etc. in the electrodeposition conditions should be investigated

-        -  And further, the dependence of the capacitance on the foam structure should be investigated.

Author Response

Expressing our thanks for the comments and suggestions of the Reviewer 2 we would like to give the following explanations.

Comment 1.   How is it possible to regulate the porosity of the metal foam? Which parameter of electrodeposition affects it?

Response: The pore formation depends on a complex process, involving nucleation growth and detachment of H2 bubbles during Cu electrodeposition. The surface pore size increases with deposition time and higher deposition current. Meanwhile, according to the reference [25], the reduction in the pore size (what is desirable in order to increase the surface area) may be achieved by adding bubble stabilizers (acetic acid), while the decrease in the branch size in the foam may be achieved by addition of chloride ions in the deposition solution. Meanwhile, for the achieving the objective of our study, we have picked one Cu 3D electrode, which was obtained under the certain electrolysis conditions. In the reviewed version we have included data on ESr of samples with different porosity. The following information was added:

  3.5. Suitability of ESr determination methods for Cu 3D structures

The results obtained suggest that in the case of plain or non-porous Cu electrodes, the electrochemically active surface area values are not affected by the determination method applied in this study. However, for Cu 3D structures, this parameter shows significant dependence on the evaluation mode. Double layer capacitance measurements result in considerably higher  values compared to the UPD technique.

In order to establish the possible reasons of this phenomenon additional measurements were carried out. The pore sizes and wall structures of the foams are tunable by adjusting the deposition conditions [25]. Considering this, Cu 3D samples were produced by varying deposition time starting from 5 up to 25 seconds and were analyzed applying Tl UPD and voltammetric double layer capacitance measurements. The surface morphology of the deposited Cu foams, except pore density and size, do not differ significantly from that of the sample presented in Figure 2a, and therefore, are not shown here. The surface roughness factors (fR) and porosity parameters of electrodeposited Cu 3D samples are listed in Table 4.

Table 4. Surface roughness factors (fR) and porosity parameters of Cu 3D electrodes deposited at 3 A cm-2 current density and various deposition durations.

Deposition duration, s

Average pore size, µm

Average pore density, cm-2

fR,

by double layer capacitance

fR,

by Tl UPD

5

7.3

3.2 107

90±12

24±4

10

16.5

1.2 107

290±19

40±9

15

21.5

8.0 106

600±26

76±10

20

25.3

4.0 106

814±29

116±12

25

28.8

3.0 106

900±38

126±13

It can be observed, that the increase in deposition time resulted in the increase of size and in reduction of density of pores in Cu 3D layer. The surface area of this type of electrodes is determined by the number and size of holes formed by detached hydrogen bubbles as well as by the wall width between them [25]. Application of both ESr determination methods revealed the increase of fR value of Cu foam with the increase of deposition duration, meanwhile, the double layer capacitance measurements yielded higher fR values compared to UPD measurements for all investigated samples.

The more detailed studies of foam formation have revealed that two types of pores are formed by the hydrogen evolution reaction during the electrodeposition process [30]. The first type are macro pores (or holes) formed by detached hydrogen bubbles, while the origin of the second type of pores is hydrogen bubbles generated at the tops of the agglomerates of Cu grains during the growth process. In addition, the pore size of foam structures increases with the distance from the substrate [30]. The analysis of the origin of the pores forming during Cu foam electrodeposition was out of scope of this study, however, it can be supposed, that with the growth of Cu 3D layer the inner or spatial structure of it becomes more and more complicated and may be the principal reason of the fR variation observed applying different ESr determination methods. Besides, the minimal 3 times difference between fR values was observed for the samples deposited at 5 s, while for all the successive ones more than 7 times difference in fR was found. The latter issues are caused most probably by some limitations of UPD-based ESr determination method.

Despite the fact that the maximum surface coverage of porous Cu electrode by Tl and Pb adatoms in UPD layer was achieved (Figure 5), it was presumed that formation of continuous and integral monolayer did not occur due to the complex spatial structure of the samples. Consequently, significantly lower  values for Cu foams were obtained. Therefore, the double layer capacitance measurements are recommended for evaluation of the electrochemically active surface area of porous Cu 3D structures.

Comment 2. The dependence of the pore size etc. in the electrodeposition conditions should be investigated.

Response: The origin of the pores during Cu foam electrodeposition was out of scope of this study, however taking in account Reviewers comments regarding the influence of porosity of Cu 3D structures we added a new subsection 3.5. Suitability of ESr determination methods for Cu 3D structures based on the results, obtained for Cu 3D structures with different porosity parameters.

Comment 3.  And further, the dependence of the capacitance on the foam structure should be investigated.

Response: Additional results regarding the SEr values of Cu foams with the different structure were obtained and the corresponding analysis was added to the manuscript in the form of a new subsection 3.5. Suitability of ESr determination methods for Cu 3D structures.

Reviewer 3 Report

The article corresponds to the subject of the journal. It is devoted to comparing the results of using various experimental techniques for studying porous copper films. The article is overloaded with a very large amount of data on the results of various measurements that are directly in its text. It would make sense to group them in the form of tables.

The results of measurements of the surface roughness factor of the same sample by different methods presented in the work differ by more than an order of magnitude (from 80 to 986). Which of the methods used can be recommended for measurements of real samples for electrocatalysis is unclear from the text of the article. The article should be revised to include a discussion of this issue. It is desirable to carry out a more detailed analysis of the data available in the literature on the applicability of various methods for measuring roughness in electrocatalytic experiments.

The following comments can be made on the text of the article:

1. It is necessary to indicate from which experimental data and how the value 2.0x10(4) cm-2 of the average pore density in the sample was obtained.

2. The surface topography data shown in Figure 1 containing 6 or more significant digits looks strange. It is necessary to more clearly indicate the dimension of the units of measurement in the given pictures.

3. The data presented in figures 1b and 2b do not agree well. In Figure 1b, the roughness is over 100 microns, while in Figure 2b it is much smaller.

4. The UPD procedure using Tl2SO4 and PbCl2 should be justified in more detail. Is supermonolayer filling of the surface possible in these experiments?

The results presented in the paper are interesting, but require a more detailed analysis of the reasons for the observed differences in the results when using different methods for measuring roughness.

Author Response

Expressing our thanks for the comments and suggestions of the Reviewer 3 we would like to give the following explanations.

Comment.1 The article is overloaded with a very large amount of data on the results of various measurements that are directly in its text. It would make sense to group them in the form of tables.

Response: All principal results obtained in this study are listed in three Tables (and now additional one with a new additional results) and we assume that there is no need to add new Tables. In order to reduce the amount of presented data we have deleted in the line 269 (reviewed version) “which ranged between 80±2 and 116±2”, as these values of were mentioned previously in Table 2. 

Comment 2. The results of measurements of the surface roughness factor of the same sample by different methods presented in the work differ by more than an order of magnitude (from 80 to 986). Which of the methods used can be recommended for measurements of real samples for electrocatalysis is unclear from the text of the article. The article should be revised to include a discussion of this issue.

Response. We accept the comment and the discussion regarding the possible reasons why different evaluation methods yielded different results was added to the manuscript in the form of subsection 3.5.

3.5. Suitability of ESr determination methods for Cu 3D structures

The results obtained suggest that in the case of plain or non-porous Cu electrodes, the electrochemically active surface area values are not affected by the determination method applied in this study. However, for Cu 3D structures, this parameter shows significant dependence on the evaluation mode. Double layer capacitance measurements result in considerably higher  values compared to the UPD technique.

In order to establish the possible reasons of this phenomenon additional measurements were carried out. The pore sizes and wall structures of the foams are tunable by adjusting the deposition conditions [25]. Considering this, Cu 3D samples were produced by varying deposition time starting from 5 up to 25 seconds and were analyzed applying Tl UPD and voltammetric double layer capacitance measurements. The surface morphology of the deposited Cu foams, except pore density and size, do not differ significantly from that of the sample presented in Figure 2a, and therefore, are not shown here. The surface roughness factors (fR) and porosity parameters of electrodeposited Cu 3D samples are listed in Table 4.

Table 4. Surface roughness factors (fR) and porosity parameters of Cu 3D electrodes deposited at 3 A cm-2 current density and various deposition durations.

Deposition duration, s

Average pore size, µm

Average pore density, cm-2

fR,

by double layer capacitance

fR,

by Tl UPD

5

7.3

3.2 107

90±12

24±4

10

16.5

1.2 107

290±19

40±9

15

21.5

8.0 106

600±26

76±10

20

25.3

4.0 106

814±29

116±12

25

28.8

3.0 106

900±38

126±13

It can be observed, that the increase in deposition time resulted in the increase of size and in reduction of density of pores in Cu 3D layer. The surface area of this type of electrodes is determined by the number and size of holes formed by detached hydrogen bubbles as well as by the wall width between them [25]. Application of both ESr determination methods revealed the increase of fR value of Cu foam with the increase of deposition duration, meanwhile, the double layer capacitance measurements yielded higher fR values compared to UPD measurements for all investigated samples.

The more detailed studies of foam formation have revealed that two types of pores are formed by the hydrogen evolution reaction during the electrodeposition process [30]. The first type are macro pores (or holes) formed by detached hydrogen bubbles, while the origin of the second type of pores is hydrogen bubbles generated at the tops of the agglomerates of Cu grains during the growth process. In addition, the pore size of foam structures increases with the distance from the substrate [30]. The analysis of the origin of the pores forming during Cu foam electrodeposition was out of scope of this study, however, it can be supposed, that with the growth of Cu 3D layer the inner or spatial structure of it becomes more and more complicated and may be the principal reason of the fR variation observed applying different ESr determination methods. Besides, the minimal 3 times difference between fR values was observed for the samples deposited at 5 s, while for all the successive ones more than 7 times difference in fR was found. The latter issues are caused most probably by some limitations of UPD-based ESr determination method.

Despite the fact that the maximum surface coverage of porous Cu electrode by Tl and Pb adatoms in UPD layer was achieved (Figure 5), it was presumed that formation of continuous and integral monolayer did not occur due to the complex spatial structure of the samples. Consequently, significantly lower  values for Cu foams were obtained. Therefore, the double layer capacitance measurements are recommended for evaluation of the electrochemically active surface area of porous Cu 3D structures.

Comment 3. It is desirable to carry out a more detailed analysis of the data available in the literature on the applicability of various methods for measuring roughness in electrocatalytic experiments.

Response: We accept the comment and the following information was added to the text (starting  from line 76 new version)

The method of monolayer formation is claimed to be more sensitive than those based on double layer charging, since the charge consumed in UPD is, as a rule, by an order of magnitude higher [7]. However, the latter method possesses several restrictions related to the facts that the identification of the end point for the metal adsorption may be uncertain, the surface distribution of the UPD species may be unknown, the adatom deposition may occur with partial charge transfer, the new phase formation may result in more condensed monolayers, multilayers or cluster growth [7]. In spite of the mentioned possible limitations UPD method is widely used for ESr evaluation of Cu samples. Meanwhile, recently Pb UPD process was applied for  evaluation of non-porous [11,16–19] and porous Cu [5] structures more frequently than Tl UPD [14,15].

Monolayer oxidation of Cu in an alkaline solution was used recently by different scholars for ESr determination of porous and non-porous Cu electrodes [3,5]. This method is applicable to metals showing the distinct regions of oxide monolayer formation and reduction [7], what is not always the case for Cu samples [16]. The oxidation of Cu in alkaline media under applied potential results in formation of different oxides and hydroxides. During oxidation scan the first current peak corresponds to the formation of Cu2O and according to [5] for the small range of OH- molarity (0.1-1 M) and scan rate (50-100 mVs-1) it can be assumed that the oxidation of Cu surface leads to the formation of a monolayer film. However, our previous studies have indicated that under the mentioned conditions Cu2O formation on the metal surface does not end with the monolayer formation, in addition this process is an irreversible one, which causes surface morphology changes and therefore is not suitable for Cu ESr determination [16].

Comment 4. It is necessary to indicate from which experimental data and how the value 2.0x10(4) cm-2 of the average pore density in the sample was obtained.

The average pore density was determined from visual analysis of the SEM images, with the lower magnification (100x) and hence large area was evaluated. One of the images is presented below.

Fig. SEM image of Cu 3D electrode.

With the SEM image presented in the manuscript we intended to expose not only the presence of pores, but also the ramified (dendritic) walls and highly developed surface area. The information regarding pore analysis was added to the text stating from 161 line (new version): Cu 3D layer porosity was evaluated applying visual analysis of SEM images.

Comment 5. The surface topography data shown in Figure 1 containing 6 or more significant digits looks strange. It is necessary to more clearly indicate the dimension of the units of measurement in the given pictures.

Response. We accept the comment regarding number of digits, however we have no possibility to change it. The images in Figure 1 were increased.

Comment 6. The data presented in figures 1b and 2b do not agree well. In Figure 1b, the roughness is over 100 microns, while in Figure 2b it is much smaller.

Response. The number indicated in the Figure 1b reflects the thickness of the Cu 3D layer, but not the roughness.

Comment 7. The UPD procedure using Tl2SO4 and PbCl2 should be justified in more detail. Is supermonolayer filling of the surface possible in these experiments?

Response. The peculiarities of the Tl and Pb UPD process on the Cu 3D samples are discussed in the form of subsection 3.5, which was added to the text (is already listed).

Comment 8. The results presented in the paper are interesting, but require a more detailed analysis of the reasons for the observed differences in the results when using different methods for measuring roughness.

Response. The discussion regarding the reasons of the observed differences in ESr values obtained by different methods is presented in subsection 3.5.

The Abstract was appended with the following sentence:

The complex spatial structure of Cu 3D layer hinders the formation of a continuous monolayer during the UPD process, being the principal reason of the observed differences. (line 24)

Reviewer 4 Report

The quality of the revised manuscript is improved enough to be published in this journal. I recommend this manuscript to be published in this form.

Round 2

Reviewer 3 Report

Necessary corrections and clarifications have been made to the article. As presented, it may be published